# Recent Advances and Current Trends in Transmission Tomographic Diffraction Microscopy

**DOI:** 10.3390/s24051594

**Published:** 2024-02-29

**Authors:** Nicolas Verrier, Matthieu Debailleul, Olivier Haeberlé

**Affiliations:** Institut Recherche en Informatique, Mathématiques, Automatique et Signal (IRIMAS UR UHA 7499), Université de Haute-Alsace, IUT Mulhouse, 61 rue Albert Camus, 68093 Mulhouse, France; matthieu.debailleul@uha.fr (M.D.); olivier.haeberle@uha.fr (O.H.)

**Keywords:** holographic microscopy, tomography, data reconstruction, polarimetric/vectorial imaging, multiple scattering, holography, diffraction, fourier optics

## Abstract

Optical microscopy techniques are among the most used methods in biomedical sample characterization. In their more advanced realization, optical microscopes demonstrate resolution down to the nanometric scale. These methods rely on the use of fluorescent sample labeling in order to break the diffraction limit. However, fluorescent molecules’ phototoxicity or photobleaching is not always compatible with the investigated samples. To overcome this limitation, quantitative phase imaging techniques have been proposed. Among these, holographic imaging has demonstrated its ability to image living microscopic samples without staining. However, for a 3D assessment of samples, tomographic acquisitions are needed. Tomographic Diffraction Microscopy (TDM) combines holographic acquisitions with tomographic reconstructions. Relying on a 3D synthetic aperture process, TDM allows for 3D quantitative measurements of the complex refractive index of the investigated sample. Since its initial proposition by Emil Wolf in 1969, the concept of TDM has found a lot of applications and has become one of the hot topics in biomedical imaging. This review focuses on recent achievements in TDM development. Current trends and perspectives of the technique are also discussed.

## 1. Introduction

Optical microscopyis an essential tool for biomedical sample characterization. Historically limited by light diffraction, optical microscopy was the room for great research efforts leading to super-resolution techniques, enabling structural imaging with a resolution down to a few nanometers [1,2,3,4]. However, these super-resolved methods rely on the use of the fluorescent labeling of the investigated sample, which can induce photobleaching and phototoxicity or can even interfere with the measured information [5,6,7].

Quantitative Phase Imaging (QPI) can be envisaged to circumvent these issues [8]. Several techniques like Fourier ptychography [9], short coherence interferometry [10], or Digital Holographic Microscopy (DHM) [11] have already been demonstrated. In the remainder of this article, we focus on the extraction of phase information considering DHM.

DHM is based on the holographic concept proposed by Gabor in 1948 [12]. Instead of recording the image of the investigated object, one records the interference between a reference field (part of the light, which does not encounter the object) and the object field (part of the light that is scattered by the object). Recording was initially performed using high-resolution photographic plates. With the development and the democratization of digital sensors, photographic plates were replaced by imaging sensors for most of the current applications [13]. In this case, the analogue reconstruction step, consisting in positioning the hologram back in the illumination beam, is replaced by digital methods [14,15,16]. It should, however, be noted that, in its original implementation, holography suffers from the so-called “twin-image” noise, resulting in a superimposition of the reconstructed object with its out-of-focus image.

This issue was tackled when Leith and Upatnieks introduced the off-axis implementation of holographic acquisitions [17]. Here, the hologram is spatially modulated due to the off-axis interference, leading to a direct separation of real and twin-image information in the Fourier space. Coupling off-axis acquisitions with Fourier space filtering, therefore, makes it possible to obtain the amplitude and phase of the object field from a single holographic acquisition [18]. Phase-shifting interferometry has also been demonstrated as a means to extract the amplitude and phase of the object field [19]. Instead of introducing a spatial modulation, the reference beam is phase-shifted with a constant phase step. Combining several acquisitions allows for the suppression of the unwanted terms in the hologram distribution, thus allowing for the complex object signal extraction. However, even if conventional reconstruction methods allow us to obtain a 3D information about both the amplitude and phase of the field, the axial resolution is lost. As a matter of fact, information is integrated over the depth of field of the microscope’s objective [20].

The axial resolution can be brought back through tomographic acquisitions, as theorized by Emil Wolf in 1969 [21]. The framework proposed by Wolf makes it possible, through the combination of 2D holographic acquisitions, to reconstruct the three-dimensional distribution of the complex refractive index of the investigated sample, while accounting for the diffraction of light at the microscopic scale. The technique, however, needs heavy computational means, which were not available at that time. The initial implementation of the technique was reported by the team of S. Kawata in Japan [22,23,24]. Nevertheless, the data reconstruction needed state-of-the-art supercomputers to yield effective results, which limited the broader adoption of TDM. A reinterpretation of Wolf’s original article made it possible for V. Lauer to build a TDM experiment relying on the use of a personal computers for data acquisition and processing [25]. Starting from this point, various implementations were proposed [26,27,28,29,30,31,32]. Note that the technique has been presented under various names: interferometric synthetic aperture microscopy, index of refraction tomography, tomographic diffraction microscopy, tomographic phase microscopy, phase tomography, optical diffraction tomography, holographic tomography, holotomography, etc. Lateral resolutions better than 100 nm have been demonstrated [32]; however, the axial resolution is still limited by the “missing-cone” problem, well known in full-field microscopy and illumination scanning tomographic configurations [33]. It should be noted that similar issues can be pointed out in sample rotation tomographic configurations [34]. Combining both illumination scanning and sample rotation is one of the solutions to this missing frequencies problem [35]. Relying on this principle, an enhanced axial resolution TDM [36] and an isotropic resolution TDM [37] have been demonstrated.

TDM is a mature technology and still an active field of research. Commercial devices are, nowadays, available [38,39,40]. This review discusses TDM theory and presents the most recent results involving either improvement in implementation, modification in the reconstruction methods, or advances in the image formation models. We also discuss the future of the technique and the answers that TDM will be able to bring to the present hot topics in the optical characterization of biological samples.

## 2. Tomographic Diffraction Microscopy

This section is devoted to TDM principles. As TDM combines QPI and tomographic acquisitions within a 3D synthetic aperture scheme, both aspects are derived.

### 2.1. General Principles: Helmholtz Equation and First-Order Born Approximation

Let us consider an object with a refractive index distribution $n\left(r\right)$ immersed in a medium whose refractive index is $n_{\mathrm{imm}}$. Let $k_{v}=2\pi /\lambda_{v}$ be the norm of the wavevector in vacuum. The wavevector in the immersion medium is, therefore, written as follows:(1)$$k_{\mathrm{imm}}=k_{v}n_{\mathrm{imm}}=\frac{2\pi n_{\mathrm{imm}}}{\lambda_{v}},$$
and
(2)$$k\left(r\right)=k_{v}n\left(r\right),$$
for propagation in the sample of refractive index $n\left(r\right)$. Using an imaging sensor makes it possible to detect part of the “total” field $u_{t}\left(r\right)$ containing information about both the field scattered by the object $u_{s}\left(r\right)$ and the illumination field $u_{i}\left(r\right)$, so that we can write the following:(3)$$u_{t}\left(r\right)=u_{s}\left(r\right)+u_{i}\left(r\right).$$
The illumination field $u_{i}\left(r\right)$ is a solution of the homogeneous Helmholtz equation, derived from Maxwell equations, and is given by the following:(4)$$\left(\nabla^{2}+k_{\mathrm{imm}}^{2}\right)u_{i}\left(r\right)=0.$$
The total field $u_{t}\left(r\right)$, given by Equation (Equation 3), is also a solution of the homogeneous Helmholtz equation:(5)$$\left[\nabla^{2}+k{\left(r\right)}^{2}\right]u_{i}\left(r\right)=0.$$
Combining Equations (Equation 4) and (Equation 5) using Equation (Equation 3) leads to the expression of the total field as a function of the scattered field:(6)$$\left(\nabla^{2}+k_{\mathrm{imm}}^{2}\right)u_{s}\left(r\right)=-k_{v}^{2}\left[n{\left(r\right)}^{2}-n_{\mathrm{imm}}^{2}\right]u_{t}\left(r\right).$$
Factorizing by $n_{\mathrm{imm}}^{2}$ in the right-hand side of Equation (Equation 6) makes it possible to express the scattered field as a function of the object’s scattering potential $V\left(r\right)$:(7)$$\left(\nabla^{2}+k_{\mathrm{imm}}^{2}\right)u_{s}\left(r\right)=-k_{imm}^{2}\left[\frac{n{\left(r\right)}^{2}}{n_{\mathrm{imm}}^{2}}-1\right]u_{t}\left(r\right)=-V\left(r\right)u_{t}\left(r\right).$$
The solution of Equation (Equation 7) is classically obtained by introducing Green’s function $G\left(r\right)$ defined by the following:(8)$$\left(\nabla^{2}+k_{\mathrm{imm}}^{2}\right)G\left(r\right)=\delta \left(r\right),$$
with $\delta \left(r\right)$ being the Dirac distribution. This methods allows for expressing Equation (Equation 7) as the following convolution product:(9)$$u_{s}\left(r\right)=\int_{V}V\left(r^{\prime }\right)u_{t}\left(r^{\prime }\right)G\left(r-r^{\prime }\right)dr^{\prime },$$
where $G\left(r-r^{\prime }\right)$ is the source of a spherical wave propagating in a medium of refractive index $n_{\mathrm{imm}}$:(10)$$G\left(r\right)=\frac{1}{4\pi }\frac{e^{ik_{imm}\left|r\right|}}{\left|r\right|}.$$
However, determining $V\left(r\right)$ from the scattering field measurement is challenging as $u_{s}\left(r\right)$ is present in the two sides of Equation (Equation 9). One possibility to circumvent this difficulty is to consider a weakly scattering object, thus simplifying Equation (Equation 3):(11)$$u_{t}\left(r\right)=u_{s}\left(r\right)+u_{i}\left(r\right)\approx u_{i}\left(r\right).$$
This constitutes the so-called first-order Born approximation. Within this framework, Equation (Equation 9) is written as follows:(12)$$u_{s}\left(r\right)=\int_{V}V\left(r^{\prime }\right)u_{i}\left(r^{\prime }\right)G\left(r-r^{\prime }\right)dr^{\prime },$$
This imaging method can, therefore, be seen as a 3D linear shift-invariant filtering operation, which not only provides a powerful analysis tool, but also permits us to compare it with other linear systems [41,42]. As we shall see, its simple geometric interpretation also provides an elegant illustration of the TDM aperture synthesis process. A Fourier space interpretation of Equation (Equation 12) is proposed in Figure 1. The incident wavefield, represented by vector $k_{i}$ (in red), is scattered by the object. The scattered field $u_{s}\left(r\right)$ is associated with the set of vectors $\left\{k_{d}\right\}$ (in green). The Ewald sphere (black) represents the domain within which light is scattered. Its radius is given by the following:(13)$$R_{\mathrm{Ewald}}=\frac{2\pi n_{\mathrm{imm}}}{\lambda_{v}}.$$

To complete the geometrical interpretation, we go back to Equation (Equation 12), decompose the illumination field on the Green function’s basis, and express the Fourier transform of $V\left(r\right)$ as a function of the scattered field spectrum:(14)$$\tilde{V}\left(k_{ox},k_{oy},k_{oz}\right)=\frac{ik_{sz}}{\pi }e^{-ik_{sz}z}{\tilde{u}}_{d}\left(k_{sx},k_{sy}\right),$$
where $k_{sx,y,z}$ are the coordinates of the scattered wavevectors $k_{s}$ and $k_{ox,y,z}$ the coordinates of the object wavevector $k_{o}$. Considering the first Born approximation, both vector sets can be linked by the following:(15)$$k_{o}=k_{s}-k_{i}.$$
Equations (Equation 14) and (Equation 15) show that the 3D spectrum of the object can be obtained from 2D measurements of the diffracted field. The coordinates $k_{sx,y}$ correspond to the back focal plane of the collection objective (equivalent to the spectrum of the scattered field), while the illumination coordinates $k_{ix,y}$ correspond to the specular spot in the same plane. The axial coordinates of both vectors are finally obtained by projecting the respective coordinates on the Ewald sphere so that the following holds:(16)$$k_{{i,s}_{z}}=\sqrt{R_{\mathrm{Ewald}}^{2}-k_{{i,s}_{x}}^{2}-k_{{i,s}_{y}}^{2}}$$

However, as it can be noticed from Figure 1, it is not possible to acquire the full Ewald sphere with a unique acquisition. The wavefield detection can either be performed in the transmission or in the reflection configuration. Moreover, microscope objectives have a limited numerical aperture (NA), limiting the norm of the wavevectors that can be acquired to $k_{\max}$:(17)$$k_{\max}=\frac{2\pi \mathrm{NA}}{\lambda_{v}}.$$
Therefore, illumination and scattered wavevectors’ coordinates should fulfill the following:(18)$$k_{{i,s}_{x,y}}\leq k_{\max},$$
which forms the so-called McCutchen pupil [43], as illustrated by the orange (reflection) and purple caps of the sphere on Figure 1, respectively. It should be noted that, within the Born approximation, the scattered field $u_{s}\left(r\right)$ scales as the cosine of the phase. Therefore, the obtained phase is determined in the $]-\pi ;\pi [$ interval, which is not compatible with objects thicker than $\lambda_{v}$.

### 2.2. General Principles: Helmholtz Equation and Rytov Approximation

The Rytov formalism has been proposed to overcome the limitations of the first Born approximation in terms of sample thickness. Here, the total field $u_{t}\left(r\right)$ is expressed as a complex phase $\varphi_{t}\left(r\right)$ [44]:(19)$$u_{t}\left(r\right)=e^{\varphi_{t}\left(r\right)}=u_{s}\left(r\right)+u_{i}\left(r\right),$$
where $u_{i}\left(r\right)=e^{\varphi_{i}\left(r\right)}$. The total field is also a solution to the homogeneous Helmholtz equation.

Let $\varphi_{R}\left(r\right)$ be the complex phase associated with the Rytov; under the assumption that $\varphi_{R}\left(r\right)\approx \varphi_{s}\left(r\right)$, the latter can be expressed as follows:(20)$$\varphi_{R}\left(r\right)=\ln\left(\frac{u_{t}\left(r\right)}{u_{i}\left(r\right)}\right)=\ln\left(\frac{\alpha_{t}\left(r\right)}{\alpha_{i}\left(r\right)}\right)+i\left[\Phi_{t}\left(r\right)-\Phi_{i}\left(r\right)\right],$$
where $\alpha_{t,i}$ are the amplitudes of the measured and illumination fields, respectively, while $\Phi_{t,i}$ denote their respective phases. Reconstruction in the Rytov approximation, therefore, generally requires a phase unwrapping procedure. Both approaches are, in fact, linked. Indeed, it has been proven that [44] the following holds:(21)$$\varphi_{R}\left(r\right)=\frac{u_{s}^{\left(B\right)}\left(r\right)}{u_{i}\left(r\right)},$$
where $u_{s}^{\left(B\right)}\left(r\right)$ is the scattered field calculated under the first-order Born approximation. Therefore, processing and reconstruction algorithms are common to both approaches. However, the range of validity is quite different: reconstructions under the Rytov approximation are more robust with thick objects [45], and the general consensus is to use the Rytov approximation in biological applications.

### 2.3. 3D Aperture Synthesis

As mentioned earlier, only a portion of the Ewald sphere is effectively collected. An enhancement of the spatial frequency content of the acquisition can be realized considering a 3D aperture synthesis scheme. It aims at combining, in Fourier space, acquisitions with variable object illumination conditions. Three common approaches can be envisaged for this task:Scanning the illumination over the object;Rotating the object within a fixed illumination;Varying the illumination wavelength.

These approaches are detailed in Figure 2. Note that here, we only focus on transmission TDM, the principles remaining the same for reflection TDM. The collected spectrum, i.e., the set of acquired $\left\{k_{s}\right\}$ wavevectors, is projected onto a cap of sphere using Equation (Equation 16), as illustrated by Figure 2a. Then, according to Equations (Equation 14) and (Equation 15), the cap of sphere is reallocated in the Fourier space as shown in Figure 2b. The main difference between the three methods relies in the way the acquired content is reallocated. It should be noted that the same Optical Transfer Function (OTF) construction mechanisms can be applied to reflection microscopy techniques, allowing us to represent, for instance, image formation in interferometric microscopy, or Optical Coherence Tomography (OCT) [42,46].

#### 2.3.1. 3D Aperture Synthesis with Illumination Sweep

In this configuration, the object is illuminated by a plane wave with a variable $k_{i}$. As the sample remains fixed, this technique is particularly stable mechanically, resolving some of the problems encountered with sample rotation. Motorized mirrors, galvanometric scanners, or Digital Micromirror Devices (DMDs), are routinely used for this task. The 3D spectrum is then built by subtracting the $k_{i}$ contribution to the acquired spectrum (set of $\left\{k_{s}\right\}$ vectors). The obtained Optical Transfer Function (OTF), as well as two orthogonal cuts in the $\left(x,y\right)$ and $\left(x,z\right)$ planes, is depicted in Figure 2c. One can notice that compared to holography in Figure 2b, the frequency content is twice as large in $\left(x,y\right)$. The axial resolution is no longer limited to the thickness of the cap of sphere but remains lower than the lateral one. Finally, one can notice a missing spatial frequency region. This is the so called “missing cone”, which is a phenomenon commonly occurring in full-field transmission optical microscopy [47,48].

#### 2.3.2. 3D Aperture Synthesis with Sample Rotation

Here, the illumination vector $k_{i}$ is fixed, and the object is rotating. As a rotation of the object is equivalent to a rotation in the Fourier space, it is possible to reallocate the resulting spectra in the OTF proposed in Figure 2d. The sample rotation can be performed through a mechanical means [37,49,50], considering the sample’s self-motion in a confined environment [51,52], or considering optical trapping techniques [53]. Sample rotation gives an almost isotropic resolution, and with a maximal theoretical gain of $\sqrt{2}$ in terms of spatial frequency support [54,55,56], compared to holography. However, frequency support remains undefined along the axis of rotation of the sample, with the final spectrum exhibiting a so-called “missing apple core” around this axis [34]. In addition, this method requires a high number of angles in order to properly sample the Fourier space, making registration between each hologram critical, possibly increasing sources of error at these scales.

#### 2.3.3. 3D Aperture Synthesis with Illumination Wavelength Variation

In this situation, both the object and the direction of the illumination wavevector $k_{i}$ are fixed. As shown by Equations (Equation 13) and (Equation 17), for a given microscope objective, the Ewald sphere radius, and, therefore, the maximal accessible spatial frequencies, are inversely proportional to the illumination wavelength in vacuum $\lambda_{0}$. Thus, taking several acquisitions at different wavelengths makes it possible to extend the frequency support [57,58,59]. One can, in some configurations, also use white-light illumination [60,61]. As illustrated in Figure 2e, the frequency support is enhanced, but in this case, the improvement is not as large as compared to the other synthesis methods.

#### 2.3.4. 3D Aperture Synthesis with Combined Approaches

None of the presented approaches succeed in completely filling the OTF: illumination scanning results in a missing region along the light propagation axis, while missing frequencies can be noticed along the rotation axis when sample rotation is considered. However, what is interesting here is that these missing regions are not along the same axes. Therefore, combining both approaches should be beneficial. A theoretical demonstration of this concept was presented by Vertu in 2011 [35], and recently, it was experimentally realized [36,37,62]. The OTFs obtained through these approaches are presented in Figure 3, with a full rotation of the sample combined with an illumination sweep as shown in Figure 3a [35,36], and with a complete illumination sweep for a few sample rotation angles as shown in Figure 3b [35,37,62].

#### 2.3.5. Examples of Achievement

In terms of the experimental implementation of the technique, as well as achievements by various groups working in the field, the interested reader is referred to Refs. [8,63,64,65,66,67,68,69]. Figure 4 gives typical examples of realization with these various approaches. On the far left is an image of a neural network obtained with a 3D aperture synthesis with an illumination sweep, adapted from Figure 2 of Ref. [32]. In this study, a 2D Sparrow resolution of about 75 nm was achieved, working at a 405 nm illumination and using two N.A. = 1.4 numerical objectives as a condenser and an imaging objective. A sub-100 nm lateral resolution was obtained when imaging a *Thalassiosira pseudonana* diatom frustule. These represent the highest obtained lateral resolutions in TDM.

The second image, Figure 6 from Ref. [70], depicts the relative refractive index distribution in an HT-1080 cell, obtained with a 3D aperture synthesis with sample rotation. In this experiment, cells were inserted inside a hollow fiber, where they grew in a similar way as in a Petri dish. The hollow fiber was rotated with a high-precision rotation stage, under the objective of a self-interference Digital Holographic Microscope (DHM), working at 532 nm, and using a 20×, N.A. = 0.4 long working distance (10 mm) objective. An isotropic subcellular resolution was demonstrated.

In order to obtain the highest-quality images, combined illumination sweeping and sample rotation can be used. This was achieved by [37] et al. using a 1.4 NA illumination/detection system working at 633 nm and attaching the observed specimens to an optical fiber, used as a rotating sample holder. The image on the right shows Figure 4 from [37], which depicts a Betula pollen grain. This setup achieved a 3D isotropic resolution of about 180 nm. This image also illustrated the interest in imaging both refractive and absorptive components. Here, images (c)–(e) clearly show that absorption is mostly located on the inner wall of the pollen envelope.

Finally, the far-right image illustrates a 3D aperture synthesis in the so-called white-light tomography. Figure 4 of Ref. [59] is that of HT29 cells in false-colour rendering. A 350 nm lateral and a 900 nm axial resolution have been obtained, lower than with the previous approaches, but still capable of revealing subcellular structures. The main advantage of this technique is that the phase imaging module is an add-on, which can be attached to a regular optical transmission microscope, contrary to the previous techniques requiring specific setups.

### 2.4. Implementation

In order to perform refractive index-resolved TDM measurements, having access to the phase of the light diffracted by the sample is mandatory. The phase can be obtained considering constrained reconstruction algorithms [71]. However, the main TDM set-ups are based on DHM configurations, as illustrated in Figure 5.

The illustrated DHM is based an a Mach–Zehnder interferometer. Light emitted by a laser is split into two distinct arms: the first one acts as a reference field (red in Figure 5), while the second illuminates the investigated sample (blue) under a Köhler illumination. In other words, the optical system formed by TL_1_, the condenser, the microscope objective, and TL_2_ is an afocal device. With this configuration, the investigated sample is illuminated by a collimated wave, which allows the use of conventional reconstruction algorithms for further data processing [21]. Please note that in Figure 5, the condenser is a microscope objective, as, for example, in experiments described in [32,37,72,73]. Here, TL_1_ enables light collimation on the investigated sample. The same can equally be performed with a more conventional condenser, as long as the object remains illuminated by a plane wave. The light diffracted by the sample (green) is collected by an infinity-corrected microscope objective. The image of the investigated sample is formed by TL_2_. For space/frequency space bandwidth product adjustment, one can consider the addition of an afocal device formed by lenses SL_1,2_ between the TL_2_ image plane and the imaging sensor. Both the illumination and scattered fields are made to interfere in the sensor’s plane. The sensor records the intensity of the diffracted field, given by the following: (22)$$\begin{array}{cc} I\left(x,y\right) & ={\left|R\left(x,y\right)+O\left(x,y\right)\right|}^{2}\\ \; & =\underset{0\mathrm{order}}{\underbrace{{\left|R\left(x,y\right)\right|}^{2}+{\left|O\left(x,y\right)\right|}^{2}}}+\underset{+1\mathrm{order}}{\underbrace{R^{*}\left(x,y\right)O\left(x,y\right)}}+\underset{-1\mathrm{order}}{\underbrace{R\left(x,y\right)O^{*}\left(x,y\right)}}, \end{array}$$
where $R\left(x,y\right)$ is the reference field, and $O\left(x,y\right)$ is the field originating from the object arm. The extraction of both the amplitude and phase of the object field is made possible through a modulation/demodulation process applied to the slowly varying envelope of the optical field. The modulation is either spatial or temporal. Spatial modulation was initially proposed by Leith and Upatnieks [17]. Here, a spatial carrier frequency (cosine fringes) is “added” to the recorded interferogram’s intensity by making the three fields interfere at a slight angle, resulting in a separation of the three diffraction orders of Equation (Equation 22) in the Fourier space. The extraction/demodulation of the complex field $O\left(x,y\right)$ is then performed through pass-band filtering, associated with carrier compensation in the Fourier space [18]. Temporal modulation has also been proposed and demonstrated by Yamagushi [19]. This method needs the acquisition of at least three holograms acquired with a properly phase-shifted reference wave. The phase shift is routinely introduced using frequency or phase modulation, which, for instance, is introduced by a mirror coupled with a piezoelectric transducer (PZT) [19], with an electro optic modulator (EOM) [30] or an acousto optic modulator [74]. If one considers the acquisition of N holograms, a relative phase shift of 2π/N between each hologram has to be imposed. In this case, the demodulation of the N hologram sequence can be performed by calculating the following:(23)$$I_{\mathrm{dem}}\left(x,y\right)=\frac{1}{N}\sum_{k=0}^{N-1}I_{k}\left(x,y\right)e^{-i\frac{2\pi k}{N}},$$
where $I_{\mathrm{dem}}$ is the demodulated field, and $I_{k}$ is the $k^{th}$ hologram of the sequence. It should be noticed that Equation (Equation 23) is equivalent to a Fourier transform performed along the temporal axis (i.e., along the image sequence). For non-regular or random phase shifts, one can apply a Principal Component Analysis (PCA) scheme on the acquired sequence [75,76].

Holographic microscopy can, in fact, be considered as a special case of simplification of tomography, with only one angle of illumination, the better-filled OTFs of tomographic systems being obtained by modifying the illumination conditions, then performing synthetic aperture. Nevertheless, it remains a method of choice for amplitude and phase extraction. From the experimental set-up presented in Figure 5, aperture synthesis can be performed by acting on the tip/tilt mirror (or other equivalent reflective beam scanning devices) by rotating the sample within the illumination beam or by modifying the illumination wavelength.

The angular scanning of the illumination beam is the most common implementation of the technique, which is now commercially available [38,39]. To scan a focused beam in the back focal plane of the illumination optics so as to deliver plane waves impinging onto the sample, stepper mirrors [25,30,31,77,78], a fast tip-tilt membrane mirrors [37,72,79,80], galvanometric mirrors [27,81,82], or rotating prisms [23,32,83] have been used. Stepper mirrors allow for very large angular deflections but are slow and vibration-sensitive. Membrane tip-tilt mirrors and galvanometric mirrors are much faster, with the galvanometric mirrors allowing for the data acquisition of several thousands of holograms per second, corresponding to acquiring about ten 3D images per second [84] (with an ultrafast camera and with post-reconstruction). In some cases, a rotating arm [85,86] has been used, performing azimuthal angular scanning ((with respect to the optical axis) with a fixed polar angle (annular scanning [72]). The Nanolive tomographic microscope is based on this principle [38]. Conversely, scanning along the polar angle can also be used [87,88].

Spatial light modulators (SLMs) deflect a wave without mechanical motion by directly acting on the wavefront phase of the illuminating beam, effectively allowing for beam angular scanning [89,90,91,92,93], or on the detected field [94]. Such systems are, however, slower than the fastest resonant galvanometric mirrors. Another technique for mechanical movement-free beam scanning uses a matrix of micromirrors (Digital Micromirror Device or DMD). Micromirrors are binary components, which switch between two positions (deflecting or not deflecting the incident beam); thus, a micromirror matrix effectively acts as a diffraction grating, which can be used to steer the illumination [95,96,97,98]. This, however, induces some constraints: it is mandatory to filter out unwanted diffraction orders to not perturb the illumination, and the possible illumination directions are not continuous (as when using a galvanometric mirror or a tip-tilt mirror). The high-resolution tomographic microscopes developed by Tomocube, Inc. [39], exploit the benefit of this approach. The concept of structured illumination has also been adapted to tomographic microscopy [90,99,100]. Note, however, that structured illumination diffraction tomography relies on a linear process with respect to field amplitude, while fluorescence structured illumination microscopy relies on a quadratic process with respect to field amplitude; thus, the underlying image formation processes are fundamentally different [101]. Structured illumination in tomographic imaging, in fact, consists of multiplexing the illuminations, demultiplexing allowing for unmixing spatial frequencies, while in fluorescence imaging, it induces spacial frequencies spectrum extension owing to a multiplicative process in a fluorescence-intensity image space, inducing the convolution of spectra in the Fourier space.

Alternately, one can also use a collection of light sources corresponding to the desired illumination directions and electronically controlled. The first implementation of this concept in tomographic imaging was performed by Isikam et al. [102] but with a 1D scan only and also using a lensless sensor. Extended to 2D angular scanning, this approach is very similar to optical ptychography in which phase and amplitude are not measured but digitally reconstructed, as shown by Horstmeyer et al. [103].

Tomography through specimen rotation is less common but presents some advantages, the most important one being that a standard holographic/phase microscope can be used, the price to be paid being the high accuracy; it is compatible with interferometric measurements and sample rotations, which have to be performed (avoiding parasitic translations and conicity). Furthermore, in order to correctly fill the Fourier space, a large number of acquisitions is to be taken, each corresponding to an accurate rotation. In some cases, this approach is simplified when the sample itself can be directly rotated without requiring a specific preparation/manipulation. This is the case when observing optical [37,104,105,106] or textile fibers [107,108]. For biological samples, they are usually encapsulated in a microcapillary [49,56,70], which serves to ensure sample rotation while simplifying specimen rotation. Note that the microcapillary may act as a cylindrical lens, distorting both illumination by the plane wave, as well as the detected image, which requires specific image reconstruction corrections [52]. Note that these drawbacks can be eliminated when the specimen is directly manipulated through standard preparation between the glass slide and cover glass in a Petri dish, or when circulating through a microfluidic device [109]. This can be efficiently performed using optical tweezers [110], which have been adapted to perform microscopic tomography [53,111,112,113], the sole constraint being performing an optically-induced rotation not around the optical axis z but perpendicularly to this axis. Optical fibers can also be used to induce sample rotation [114,115]. Note that any phenomenon inducing a rotation of the sample can indeed be used to perform tomography with sample rotation. Dielectrophoresis (more precisely, electrorotation) [116,117,118], when an external electric field induces sample rotation because it presents electrical potential variations, as well as acoustophoresis [119] (also known as sonophoresis or phonophoresis), the acoustic manipulation of microscopic samples, have also been used in optical tomography [120].

Finally, as presented in Figure 3, combining sample rotation with illumination rotation [35] permits us to suppress the so-called “missing cone” characteristic of transmission microscope [47] as well as the residual “missing apple-core” of sample rotation tomography [34]. The first attempt to conduct this proposal was performed through imaging microfibers or attaching the sample to a micro tip used to rotate the specimen [37], but recently, optical tweezers have also been used with configurations corresponding to Figure 3a [35,36], or Figure 3b [35,62].

One of the main flaws of TDM is its lack of selectivity in the reconstructed refractive index content. As a matter of fact, different classes of structures can exhibit the same refractive information. This issue can be tackled by combining TDM with more conventional fluorescence labeling techniques [78]. Other imaging contrasts can also be investigated. Indeed, TDM allows for the sequential acquisition of the optical field along various points of view of the sample, making it possible to mimic, by numerical means, conventional routine microscopes [121].

## 3. Data Reconstruction and Multimodal Imaging

As TDM grants access to 3D information about the optical field scattered by the investigated sample, it can be used to mimic conventional microscope either in 2D or in 3D [121,122,123,124,125,126,127]. Simulation schemes are depicted in Figure 6. Compared to conventional microscopy techniques, TDM performs sequential acquisitions of the optical field. Considering 2D microscopes, the general image formation model can be summarized as follows:Holograms that will be used for simulation are selected according to the illumination angle, which is equivalent to the illumination selection performed by the condenser in a conventional microscope.Optional spectrum filtering can be envisaged. This is the numerical equivalent of the backfocal plane pupil filter that can be found in some specific microscope objectives. For instance, considering the case of the Zernike phase-contrast microscopy [128], one can multiply the following pupil filter to the hologram’s Fourier transform:
(24)$$P\left(k_{x},k_{y}\right)=\alpha \left(k_{x},k_{y}\right)e^{i\Phi \left(k_{x},k_{y}\right)},$$
which allows for both the phase shifting and attenuation of the specular illumination contribution (part of the illumination beam that does not encounter the sample). In this situation, the attenuation can be defined as follows:
(25)$$\alpha \left(k_{x},k_{y}\right)=\left\{\begin{array}{c} \alpha \;\;\mathrm{if}\;\sqrt{{\left(k_{x}-k_{i_{x}}\right)}^{2}+{\left(k_{y}-k_{i_{y}}\right)}^{2}}\leq \rho_{\mathrm{pc}}\\ 1\;\;\mathrm{elsewhere} \end{array}\right.,$$
where $\rho_{\mathrm{pc}}$ is the radius of the pupil filter, and α is the attenuation ratio. Here, the filter is centered on the specular illumination coordinate $\left(k_{i_{x}},k_{i_{y}}\right)$. The same applies to the phase shift Φ, which can be expressed as follows:
(26)$$\Phi \left(k_{x},k_{y}\right)=\left\{\begin{array}{c} \Phi \;\;\mathrm{if}\;\sqrt{{\left(k_{x}-k_{i_{x}}\right)}^{2}+{\left(k_{y}-k_{i_{y}}\right)}^{2}}\leq \rho_{\mathrm{pc}}\\ 0\;\;\mathrm{elsewhere} \end{array}\right..$$It should be noted that, in the case of the Zernike phase contrast, the condenser selects illumination angles along an annulus. Therefore, if we add all the processed holograms, the proposed filter will map an annulus in the Fourier space. This annulus is the perfect equivalent to the one existing in a phase-contrast microscope objective.The multiplexing of each processed hologram is finally performed by summing the calculated contrast intensity.

**Figure 6 sensors-24-01594-f006:**
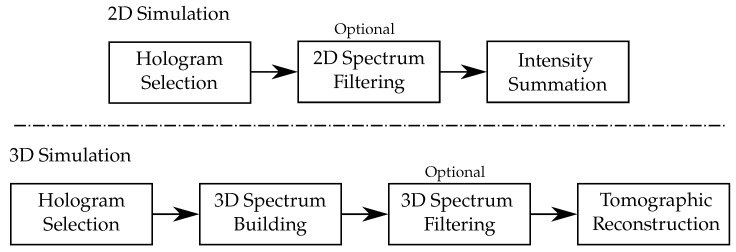
The 2D and 3D simulation principles for multimodal reconstruction.

An example of multimodal reconstruction is shown in Figure 7 [121]. Here, using the scheme developed in Figure 6, we were able to mimic the behavior of the following: (a) a brightfield, (b) a darkfield, (c) oblique illumination, (d) Rheinberg illumination, (e) negative phase contrast, and (f) Differential Interference Contrast (DIC) microscopes. More details about the simulation process can be found in Ref. [121]. The Python version of our simulation and tomographic reconstruction codes can be found in Github/PretraitementMulti.py version 0.0.1 (initial commit) [129].

It is also possible to extend the previous results to the third dimension. For this purpose, we built a specific scheme, presented in Figure 6, as an extension to the bidimensional case. Here, multiplexing is performed in the 3D Fourier space prior to backfocal plane filtering. The final step is, thus, a conventional tomographic reconstruction. The obtained results, in both lateral and axial cuts, are shown in Figure 8 for the (a,f) darkfield, (b,g) phase contrast, (c,h) DIC, and (d,i) Rheinberg illumination microscopy. Cuts of the composite RGB Rheinberg spectrum are proposed for illustration purposes (e,j).

Note also that, at least for a sample satisfying first Born approximation, the real-time 3D acquisition/computation/display of high-resolution TDM images in augmented reality becomes a real possibility. Very fast cameras [84] and powerful GPU cards [130,131,132,133,134,135] are available and have already been used for TDM. At present time, data transfer from the high-speed cameras to the GPU card may still constitute some bottleneck, but coupling with a holographic screen may, in the near future, open the path to a live display of microscopic specimens in true 3D.

Finally, precise measurements of the electromagnetic field resulting from light-sample interaction not only allow for numerically recreating image formation in any type of optical microscope (using same illumination/detection numerical aperture), but also permit us to numerically manipulate the optical properties of the sample itself. In this view, the so-called in silico clearing of the observed sample has been recently proposed to image highly scattering spheroids [136,137]. In this technique, layer-by-layer sample reconstruction allows for suppressing multiple scattering and sample-induced aberration from one layer in order to reconstruct the next layer, providing a numerical equivalent of chemical tissue clearing [138].

## 4. Advanced Reconstruction Methods

The description of the system obtained in “classical” TDM is based on the scalar Helmholtz Equation (direct problem) and on the weak scattering hypothesis (inverse problem). Under these conditions, the field is linearly related to the refractive index via the Born or Rytov approximation. The system can be described by its Optical Transfer Function (OTF) in the Fourier space, or, equivalently, by its Point Spread Function (PSF) in the image space. As explained above, it provides a set of elegant and powerful tools: it gives a simple geometric description of the system and allows for performances to be compared with other devices owing to the OTFs, such as reflection tomographic diffraction microscopy, interferometric microscopy, and OCT… [42,46].

However, these approximations can be challenged by thick samples or samples with a high refractive index contrast. Refraction modifies the illumination wave, and the plane wave hypothesis degrades with depth. Multiple scattering is ignored, which can lead to severe reconstruction problems.

Apart from the reconstruction artefacts (morphology, index or absorption values) induced by these effects, taking account of multiple scattering could, in some cases, even increase resolution beyond the classical limits. Although it remains an open debate, one explanation is that evanescent waves generated in an inhomogeneous medium are converted into propagating waves by multiple scattering. Since evanescent waves can contain subwavelength information, multiple scattering could encode a super-resolution in far-field diffraction information [139,140]. It should be noted though, that in MTD, these improvements in resolution appear to be closely dependent on the experimental configuration [141].

### 4.1. Iterative Reconstructions

Inversion methods based on non-linear models can include these effects to improve the reconstruction. To find the refractive index *n* at each pixel in a set Ω, the inversion scheme generally relies on an optimization-based method, where a cost function C(n) is minimized [142,143]:(27)$$\hat{n}=\underset{n\in \Omega }{\mathrm{argmin}}\left\{C\left(n\right)\right\}=\underset{n\in \Omega }{\mathrm{argmin}}\left\{D\left(n\right)+\tau R\left(n\right)\right\}$$

D(n) measures the fidelity of the forward model to data, generally calculated over the *P* illumination angles with an $l_{2}$ norm:(28)$$\frac{1}{P}\sum_{p=1}^{P}{∥{\hat{m}}^{p}-m^{p}∥}_{2}^{2}$$

The estimated scattered field $\hat{m}$ is calculated with the forward model and compared to the measured scattered field *m* to obtain the data fidelity term. As the problem is generally ill-posed, a regularization term R(n) is added to the data fidelity term [144] whose weight is adjusted by the parameter τ.

The direct model can be very general, avoiding the approximations of analytical approaches. It can also be formulated with intensity measurements, simplifying the experimental set-up [145,146,147]. Figure 9’s top row (Figure 6 from Ref. [145]) illustrates the superior results obtained when reconstructing a rather large sample, an entire adult *C. elegans* worm when using a multi-layer Born multiple-scattering approach, compared to simple inversion based on the first Born approximation, or a multi-slice beam propagation method.

### 4.2. Forward Models

The accuracy of the forward models is crucial and mainly limited by computational power and memory limit. Rigorous solutions of the Maxwell equations can be calculated using the FDTD or finite element methods, providing a ground truth for the simulations and testing of lighter algorithms [150,151], but their complexity is still prohibitive in iterative algorithms. In TDM, early works using advanced non-linear models (coupled dipole, contrast source inversion) were conducted in reflection TDM, with a simple geometric object and limited field of view [152]. They showed the possibility to improve a lot the reconstructions, even beyond Abbe’s limit [153].

The choice of a direct model is, therefore, often based on a compromise between accuracy and reconstruction speed.

#### 4.2.1. Multi-Layer Models

An interesting approach is to divide the object into thin slices and apply an operator that calculates the effects of diffraction and refraction using the 2D refractive index map inside this slice. The total wave at the output of one layer becomes the illumination of the next one.

The calculation of the complex field at the next layer, u(x,y,z+Δz), can be performed using the beam propagation method (BPM) [154,155], in paraxial version [142,148,156] or improved versions [150,157]: (29)$$u\left(x,y,z+\Delta z\right)\propto F_{2D}^{-1}\left\{F_{2D}\left\{u\left(x,y,z\right)\right\}\exp\left(-i\frac{k_{x}^{2}+k_{y}^{2}}{k+k_{z}}\Delta z\right)\right\}\times \exp\left(i\frac{k_{v}\Delta n\left(x,y;z\right)\Delta z}{\cos\theta_{i}}\right)$$
where $\theta_{i}$ is the illumination angle, Δz is the layer thickness, *F* is the 2D Fourier transform, and Δn(x,y;z) is the refractive index contrast.

Under a small thickness approximation, as can be seen in Equation (Equation 29), this method separates the refraction and the diffraction operators, which are successively applied in the direct (refraction) and Fourier (diffraction) spaces. This algorithm is potentially very fast when using a GPU, since the calculations are carried out in 2D and can use Fast Fourier Transform (FFT).

However, it ignores reflection and does not take into account index inhomogeneity in the diffraction operator. This last problem can be addressed with the wave propagation method (WPM) [158,159]: instead of propagating the field in the mean refractive index (first term in Equation (Equation 29)), a multiplication by the 2D (N×N) refractive map of the current layer is preformed. The price to be paid is an increased complexity ($∼N^{2}$) compared to BPM (∼NlogN), since FFT cannot be used any more. Hybrid methods using both BPM and WPM have also been proposed to alleviate the problem [160]. It should be noted that WPM also ignores reflection and evanescent waves, at least in its original form [161].

A similar multi-layer approach using the first Born approximation has been proposed to take into account both multi-scattering and backscattered waves [145]. The complex field is calculated by sequentially applying the first Born scattering to each layer of finite thickness. When this 3D model is applied to 2D thin slabs, it is worth noting that Green’s function must be modified with an obliquity factor [162,163]. In each slice, the algorithm consists of two steps:Propagation of the incident field (total field from the previous layer) with the angular spectrum method [164],Calculation of the scattered field under the first Born approximation, i.e., convolution between the modified Green function and the product of the incident field by the potential (Equation (Equation 12) on the slice thickness).

Like the BPM method, this method is also potentially fast, since these calculations are carried out with 2D FFTs.

#### 4.2.2. Full 3D Models

More advanced, and more complex, forward models are based directly on the solution of the scalar Helmholtz equation obtained from the Green’s function theorem (Equation (Equation 9)), i.e., the Lippman–Schwinger equation for the total field. This equation can be solved with iterative schemes and has been applied successfully to 2D problems [147,165]. The iterative forward scheme on 3D arrays’ computation makes them more demanding, requiring more computing power and memory than multi-slice models. Thus, some works have focused on reducing this cost in order to apply it to 3D/larger structures [166].

An alternative is to use the classical Born series, which arises naturally from the solution under the first Born approximation. In order to improve this approximation, the total field obtained under the first Born approximation can be used as the incident field in Equation (Equation 9). A recurrence relation is then defined between the different orders of scattering [167]:(30)$$u^{n+1}\left(r\right)=u_{i}\left(r\right)+\int_{V}u^{n}\left(r^{\prime }\right)f\left(r^{\prime }\right)g\left(r-r^{\prime }\right)d^{3}r^{\prime }$$

However, its practical use is limited by the convergence criteria, restricted to the case of small objects, or low scattering potential [168]. Recently, a modified Born series was proposed to achieve convergence, regardless of the size and the refractive index contrast [169] and has been used as a forward model in TDM on 3D objects [149].

The convergent Born series has been extended to solve the full wave Equation (Maxwell’s equations) [170], which opens the way for solving the most complex problems, including multi-scattering and vectorial aspects of light.

Figure 9’s middle and bottom rows (adapted from Figure 4 from Ref. [149]) demonstrate the higher-quality results obtained by Lee et al. with an inverse problem approach using modified Born series to image optically thick samples, compared to Rytov approximation and Rytov approximation with the total variation regularization.

## 5. Accounting for the Vectorial Nature of Light

In its simpler implementation, TDM relies on a scalar and paraxial resolution of the Helmholtz Equation [21]. This formalism does not make it possible to account for light polarization. In 2002, Lauer proposed a vectorial extension of the Born formalism [25], which was still limited by the paraxial propagation assumption, i.e., considering ∇∇·E=0 in the propagation equation. The vectorial and non-paraxial resolution of the Helmholtz equation has been demonstrated, leading to a complete formalism that is especially useful in situations where light propagation is no longer linear (e.g., in the presence of the Kerr effect) [171]. Approaches presented in the previous section, based on coupled dipole [141], propagating approaches [142], or Born series decomposition [169,170] can also be considered to perform reconstruction beyond the scalar approximation.

The qualitative use of polarization as a novel imaging contrast in a biological sample was recently proposed for the characterization of zebrafish embryo [172,173]. Here, conventional TDM images are acquired under two different illumination polarization directions, making it possible to fuse refractive index information (transparent part of the fish) with a pseudo-birefringence contrast (mainly associated with fish bones).

The quantitative extraction of polarization metrics has been recently demonstratted [73,150,174,175,176]. Here, considering a 2D approximated vectorial model (polarization of the sample is neglected along the light propagation axis) [73,150,175] or a full 3D-tensorial approach [176], the authors linked the reconstructed data to the Jones tensors of the sample. This is made possible at the cost of multiple tomographic acquisitions with varying illumination polarization scenarios. For instance, four TDM acquisitions are needed if one considers linearly polarized illumination. These can be achieved using a single [150] or two [175] conventional cameras. Using a Polarization Array Sensor (PAS) as an image detector allows us to divide the amount of acquisitions by a factor of 2 [73]. Here, illumination and reference beams are circularly polarized. Polarization analysis is intrinsically performed by the PAS at the cost of data demosaicking and interpolation [177]. An example of quantitative polarization measurements, adapted from Ref. [73], is presented in Figure 10. Here, a potato starch is used as a birefringent sample. One can realize that polarization information is lost on both brightfield Figure 10a and conventional TDM reconstruction Figure 10d. The well-known Maltese cross structure is made visible with a polarized light microscope (PLM) as shown in Figure 10b and confirmed with the calculated birefringence map as shown in Figure 10e. A qualitative polarization orientation can be obtained by adding a phase plate to the PLM as shown in Figure 10c, revealing a phase opposition between two consecutive branches of the Maltese cross. This is confirmed by the quantitative polarization orientation measurement (f).

As mentioned earlier, the method proposed in Ref. [73] implies image demosaicking coupled with data interpolation. It should be noted that, due to the intrinsic structure of a PAS, coupled with the holographic nature of data acquisition, removing the interpolation for the data processing workflow is equivalent to performing 3D-DIC acquisitions [178]. This is a very interesting feature, indeed making the PAS implementation of TDM a multimodal imaging technique.

## 6. Present and Future Trends

While the technique as been intensively investigated and improved, being now even commercially available, it still suffers from some limitations and/or complexities, which necessitates new approaches in view of simplifying/ruggedizing its hardware implementation, accelerating the acquisitions, expending potential applications, or improving measurement reliability, which will necessitate implementing metrological approaches to the index of refraction measurements.

### 6.1. Hardware Simplification

For image reconstructions, TDM requires few tens to few hundreds of acquisitions, depending on the acquisition technique (illumination rotation, sample rotation, or both), reconstruction algorithms (simple direct inversion or more elaborate iterative methods), as well as targeted image quality and resolution. The sequential acquisition necessary to perform synthetic aperture, by definition, slows down the image capture rate. This explains that for holographic acquisitions, off-axis setups [179] are preferred to phase-shifting approaches [19], which, while allowing for a larger field of view, require several intermediary acquisitions for each tomographic angle, even further slowing down the process. An optimal scanning pattern helps for minimizing the number of illumination angles to be used while maintaining a good image quality. Using 1-D line scanning allows for rapid acquisitions but delivers non-isotropic-resolution images, even in the x–y plane [87,180]. Star [30,77], circular [96,181], flower [31,37,78], or spiral [182] scanning patterns have been used. However, a better filling of the Fourier space is obtained with more elaborate scanning patterns [72,183,184,185] (note that the demonstration of the existence of an optimal scanning scheme, as well as its exact determination are still to be performed). The preprocessing of the holograms [52,186], or the implementation of adaptive illumination [91], for example, to correct for residual aberrations, as well as data sorting to suppress bad-quality data [187,188], also help to maintain a good image quality while trying to minimize the volume of acquisitions.

Data acquisitions, however, can be significantly accelerated by acquiring two multiplexed off-axis holograms [189], a technique which has been expanded to six-pack holography [190], and even the so-called double six-pack holography, which allows for dynamic tomography by simultaneously acquiring 12 holograms arising from 12 angles of illumination [191]. In some cases, sample properties can be used to accelerate data acquisition [192].

Snapshot tomography is another possible approach to multiplexing holograms and accelerating acquisitions [193,194,195,196]. This method, compared to the six-pack method or double six-pack holography, has the advantage of simplicity, through multiplexing the illumination using a microlens array, which translates the detection camera into many sub-apertures, each delivering an off-axis hologram corresponding to a different illumination angle. The price to be paid is, however, that each multiplexed hologram is captured with a much lower effective numerical aperture, while the six-pack approach makes optimal use of the objective numerical aperture.

Another approach to simplifying the hardware implementation of holographic tomography comes from the fact that, in its most common implementations, it requires an interferometer to record holograms, which can be sensitive to vibrations or air flows, degrading the hologram quality. Common-path holography has been intensively developed to address these issues [197]. This approach has also been adapted to tomography. However, contrary to holography, the illumination direction changes, so that it requires a descanning galvanometer in order to compensate for the illumination beam direction and properly refocus it through a pinhole to regenerate a plane reference wave [198]. This added complexity has inspired other approaches, such as performing phase-shifting holography using a Spatial Light Modulator [181] or sharing interferometry via SLM [199], gratings [86,200] or polarizing elements [201]. These methods, however, require observing sparse samples, so as to use as a reference beam a part of the illumination that has traversed an empty zone of the observed area. Alternately, one can use a wavefront sensor [202,203].

Another approach consists of recording intensity-only images, which are then numerically processed to reconstruct phase and amplitude images [71,204], to be properly recombined for tomography [205]. In such approaches, the hardware complexity of measuring both the amplitude and phase of the diffracted field is transferred into the software complexity of the numerical reconstruction of these quantities from intensity-only measurements. This explains that these approaches have long been limited by the available computer power but have experienced a growing interest in recent years [206,207,208,209], allowing for high-throughput imaging [210,211,212]. In recent years, approaches based on the transport of intensity Equation [213,214,215,216,217] or the Kramers–Kronig relations [121,218,219] have been developed, and multiple scattering [147,148,160,220] and polarization-sensitive [221] versions have also been implemented. Tomocube, Inc. [39], introduced a low-coherence light source tomographic system. For such approaches, deep learning methods to recover the phase [222,223] appear particularly promising [224,225]. Note that as for holographic tomography, the optimization of illumination in the case of partially coherent [207,226,227] and incoherent intensity tomography [228,229,230] is also of importance.

Finally, lensless tomography [88,102,231] is another technique, which permits us to avoid the use of an interferometer by directly recording the interference fringes produced by the sample deposited on, or very close to, the electronic sensor. Doing so leads to the simplest tomographic configurations that are able to perform 3-D imaging of rather large samples (e.g., *C. elegans* nematode) using LEDs for illumination and without requiring focusing optics.

It is also interesting that in some cases, tomographic acquisitions can be performed without having to steer the illumination nor control the sample rotation. This is the case for samples that can freely rotate under peculiar conditions such as when flowing through microfluidic channels. This property of self-rotation has been cleverly used to image red blood cells and diatoms [232], image human breast adenocarcinoma MCF-7 cells [233], identify cell nuclei [234], perform single-cell lipidometry [235], or study yeasts [236]. Note that, in fact, one can obtain the benefit of any phenomenon inducing an uncontrolled rotation of the observed sample [237] if one is able to accurately recover the rotation angle in order to properly reassign information for the reconstruction algorithm [238].

Figure 11 depicts three simplified tomographic systems, highlighting complementary approaches with specific advantages. Snapshot tomography [193,194,239] divides the objective numerical aperture using a microlens array to allow for simultaneously capturing many views (Figure 11, left). This approach is potentially the fastest for tomographic imaging, being only limited by the acquisition speed of the camera. Lens-free tomography (Figure 11, middle) simplifies the hardware by removing all the optical systems for acquisition (microscope objective, interferometer) and using a limited number of LED illuminations. In [240], Luo et al. proposed an extreme simplification with only four illuminations and advanced reconstruction methods, allowing for imaging very large samples with lateral dimensions in the millimeter range and a thickness of several hundreds of micrometers. Tomographic flow microscopy (Figure 11, right) allows for using a simple holographic setup for data acquisition, the diversity of illumination being provided by natural sample rotation in a microfluidic channel. The numerical difficulty for reconstruction is the correct estimation of the rotation angles, for which specific algorithms have been developed. They allow for efficient 3D image reconstruction, as illustrated by the image of a red blood cell exhibiting one-side-concavity morphological anomaly.

### 6.2. Functionalization of Tomography

The functionalization of TDM can be envisaged as the addition of a new quantitative imaging contrast to the conventional refractive index modality. For this purpose, we already discussed the “numerical functionalization” that can be achieved by modifying data reconstruction to mimic a conventional microscope. This functionalization has also been proven to be possible by acting on the experimental set-up for accounting for sample polarizability.

While the index of refraction measurements can be very precise and sensitive and provides some chemical selectivity, it is, however, rather weakly discriminating, as different species/structures can exhibit the same index of refraction when measured at single wavelengths. However, the optical index varies with wavelength, especially its absorption component. Up till now, absorption is simply neglected in the commercial implementations of TDM, as well as in the vast majority of published research, which can be considered as surprising but is linked to limited reconstruction models or simplified acquisition systems. Hyperspectral systems have been developed to overcome this limitation [241,242]. In particular, Sung [242] developed spectroscopic microtomography with a sensitivity high enough to distinguish, at the single-cell level, oxygenated from deoxygenated red blood cells. Recently, hyperspectral tomography was combined with SLM acquisitions [239,243].

Tomographic diffractive microscopy has been mostly developed in the visible range, but recently, a tomographic setup working in the near-infrared range was developed by Ossowski et al. using a tunable semiconductor laser as the light source, set from λ= 800 to 870 nm [244]. Note that such a short-coherence source necessitates the addition of an optical-path-difference adjustment module in the system. After a proper calibration, this setup is able to identify specific structures related to colon cancer in unstained histologic sections, demonstrating the interest in working with near-infrared wavelengths in tomographic approaches [244]. Using a similar approach, Juntunen et al. developed spectroscopic microtomography in the short-wave infrared range, taking advantage of the much wider illumination spectrum provided by a supercontinuum laser and the larger-depth penetration possible in the SWIR range [245] to study large samples such as human hair and sea urchin embryos in various developmental stages.

Another promising approach to improve the chemical sensitivity of holographic/tomographic imaging is based on recording phase/index changes induced by photothermal effects due to infrared absorption. It results in the so-called bond-selective imaging, which has been successfully implemented in both interferometric- [246,247] and non-interferometric diffraction tomography [248,249], achieving sub-micrometer volumetric chemical imaging in individual cells, but also in model organisms such as *C. elegans* nematodes; but the technique has also been used for material studies [250].

Apart from chemical selectivity, one can also add structural selectivity by taking advantage of harmonic generation, which is characteristic of the structural architecture of the sample. Harmonic holography was proposed about 15 years ago [251,252,253,254,255,256,257,258]. Tomographic extensions have also recently been studied [259,260,261], benefiting from the selectivity of the second harmonic generation (SHG) to specifically identify non-centrosymmetric structures within the observed sample and from the better resolution provided by synthetic aperture imaging. Its extension towards the third harmonic generation (THG) could, for example, allow for the identification of sub-micrometer-specific structures (lipidic droplets, for example [262]). Polarized SHG/THG is also sensitive to optical anisotropy; furthermore, SHG and THG often deliver complementary information about the sample [263].

In most TDM experiments, static or slowly moving objects are studied. Hugonnet et al. proposed an original approach to the visualization of 3D refractive index dynamics. Through the appropriate spatial filtering of tomographic data, they were able to study slow and fast movements of subcellular organelles and biological molecules within living cells, which should help in expanding the applications of the index of refraction imaging [264].

Other dynamic information can be exploited. As a matter of fact, the use of Doppler light broadening has already been demonstrated in the framework of digital holographic imaging for non-destructive testings [265,266], zebrafish blood flow assessment [267], or full retina quantitative blood flow imaging [268,269]. An extension to the third dimension has been demonstrated with a small number of illumination angles, considering the reconstruction of the blood flow under sparsity constrains [270,271].

Finally, the use of phase contrast agents could allow for improving specificity, but obviously at the price of having to abandon working with unlabeled samples, which is the main advantage of the technique. As for fluorescence imaging, a genetically encodable phase contrast agent has been recently proposed [272] based on gas vesicles and used as a biomolecular contrast agent easily identifiable using digital holographic microscopy.

### 6.3. Metrological Approaches

An important point of view of a wider adoption of the technique is the development of metrological approaches [273,274], so as to guarantee the validity and domain of confidence of measurements, which is especially important for material science and industrial applications. For biological investigations, the large natural variations between samples often lead to the need for averaging measurements or only studying temporal variations within the same sample; and often, relative measurements may suffice to distinguish sub-cellular compartments, for example, lipid droplets having a higher index of refraction [235,275] than their surrounding cellular medium. The situation is often different when studying manufactured samples, such as optical fibres [37,104,276,277,278], photopolymerized structures [199,279], or plastic lenses [280], for which absolute measurements are often required to precisely characterize the sample properties both in terms of the dimensions and optical indexes of refraction, for example, to optimize a fabrication process. To do so, one often works using rather simple test samples such as calibrated beads, USAF [281], or Siemens [273] test patterns, but recently, more complex, true 3-D structures have been developed [282,283] in order to precisely characterize instrumentation performances [284]. Figure 12 (Figure 8 from Ref. [284]) illustrates the differences in shapes and optical refraction index measurements about the same sample when imaged with three different tomographic systems, highlighting the importance of developing standards in the domain.

Discrepancies in shape and/or measured indexes arise from restrictions in data acquisition (missing-cone problem, anisotropic resolution, and/or sub-optimal scanning schemes), limited accuracy (e.g., when a large index of refraction differences is present within the samples, or between the sample and its immersion medium), or underlying simplifications (e.g., neglecting absorption) of reconstruction algorithms. A simple technique to avoid an excessively large index mismatch is to choose an immersion medium with an index of refraction close to that of the observed sample, which is rather easy for artificial samples, but can even be performed in some cases for biological living specimens [285].

Note, however, that for the smallest structures, close to the limits of the instrument, the strong anisotropic resolution, characteristic of all transmission microscopes, will always constitute a problem: such minute structures are indeed observed as being elongated along the optical axis, which renders any volumetric measurements problematic (if one does not make supplementary assumptions such as assuming that these structures are spherical, for example). The importance of working with isotropic-resolution images is often overlooked: as even simple quantities such as volumes cannot be accurately measured at submicrometric scales in transmission microscopy, index of refraction or species concentrations are even more difficult to estimate with precision, which necessitates new developments to improve the resolution.

### 6.4. Promising Applications

As mentioned throughout this review, TDM has already found wide application in various fields such as biomedical imaging, material science, and surface characterization.

In particular, the technique has been successfully used to study a large variety of samples in view of biological applications: red or white blood cells, hepatocyte cells, cancerous cells, neuronal cells, chromosomes, mechanisms of cell–cell or cell-to-surface adhesion, and human hairs; it has also been used to study bacteria, pollens, etc. (see articles [8,64,65,66,67,68,286,287] and references therein). In fact, all applications for which phase imaging microscopy is successful [288,289] would benefit from the superior imaging capabilities of TDM, except (at least up to now) those requiring ultrafast imaging, such as the high-resolution imaging of erythrocytes’ vibrational modes [290], for which sequential acquisitions of data remain a big hurdle.

Note that applications have been greatly boosted by the availability of commercial implementations of the technique of TDM with illumination rotation or in white-light illumination. The interested reader will find numerous examples of the applications on biological samples on the Nanolive [38], Tomocube [39], and PhiOptics [40] websites. The Nanolive system has the advantage of offering a large accessible space over the sample, while the Tomocube system allows for faster acquisitions at slightly higher resolutions. Both systems can also be upgraded with fluorescence imaging as an option. The white-light TDM of PhiOptics has been developed as a dedicated add-on module, with the advantage that this module can be fitted to a standard microscope’s body, allowing for a new imaging modality on an existing instrument.

These instruments also allow for high-throughput screening investigations, using 96-well plates, either directly added to the system (Nanolive, PhiOptics) or via a dedicated variant of the apparatus (TomoCube). Another promising approach to high-throughput screening is the so-called tomographic imaging flow cytometry [291], also developed in a holographic version [232]; while being limited to free-standing samples capable of spontaneously rolling through microfluidic channels, it offers a simple approach to studying a large number of individual cells, e.g., for lipidometric investigations [235], or to studying nanoparticle internalization by cells [292]. TDM with sample rotation is another possible approach to high-throughput screenings of cells when using a microcapillary to control cell flow and sample rotation under the objective [70]. Microfluidic channels can also be used [293,294]. However, at the present time, no such from-the-shelf system is yet available, limiting similar investigations of instrumentation development teams, while several companies offer holographic amicroscopes [295,296,297,298] or phase cameras [299], which could be used for such research.

Organoids or spheroids [300] are a hot topic in the community of optical imaging [301,302,303]. They, however, present specific difficulties in terms of size, multiple diffraction, and/or absorption, which challenges the capabilities of current tomographic imaging systems, which have mostly been developed to work at the cellular level and not at the tissue level. Three-dimensional cell cultures [88,240] present similar challenges, and so do animal models such as *C. elegans.* worm, Zebrafish, or Xenopus frogs. Promising results on samples as large as Zebrafish embryos have indeed already been obtained [102,172,231,270,271,304]. However, corresponding instruments have not yet been deployed outside instrumentation laboratories, limiting their application, but the availability of tomographic microscopy techniques adapted to such biological system would certainly trigger a blooming of applications.

While most TDM applications have been in biological research up till now, note that a growing interest appears in material sciences and micro-nanofabrication [37,104,199,276,277,278,279,280]. As mentioned in the *Metrological approaches* subsection, research in these domains is often more demanding in terms of the index of refraction calibration and absolute precision, which may explain why the technique is not as widely used as in biology; thus, applications are less common. The situation is similar for holographic microscopy and tomographic microscopy in the reflection mode, which have been less developed and are not yet commercially available (with the noticeable exception of the Reflection DHM of LynceeTec [295]).

We, however, believe that forthcoming improvements in the technique will motivate the dissemination of optical tomographic microscopy in these fields, too, provided that some specific challenges, exposed in the next section, can be addressed.

### 6.5. Current Trends and Challenges

Optical microtomography has experienced tremendous development in recent years. Many variants have been developed and tested, some having been commercially implemented. Nevertheless, several hurdles still exist, which have motivated many groups to try improving the performances of tomographic approaches even further.

As previously mentioned in the metrological approaches section, a strong limitation, when trying to quantify the volume and/or the index of refraction of the smallest sample details, comes from the anisotropic resolution of transmission optical microscopes. Combining illumination rotation and sample rotation delivers isotropic-resolution images [35,37,62], but this approach is limited to free-standing samples one can manipulate, and it is not, for example, for cells cultivated on a slide. Lauer [25] proposed adapting the so-called 4Pi configuration to tomographic approaches, performing both transmission and reflection acquisitions. Preliminary experiments have validated the concept of dual transmission and reflection acquisitions on semitransparent samples [281], but in this experiment, independent transmission and reflection reconstructions were performed because no common data between transmission and reflection OTF could be acquired to perform a synthetic aperture. Mudry et al. [305] proposed an elegant solution to this problem, with the so-called mirror-assisted tomography, which, in fact, amounts to folding the 4Pi setup onto itself, owing to the mirroring effect. Using the mirror effect allows for acquiring two transmission and two refection data set using only one objective and one camera. A simplified version has already been built [80], demonstrating that a reflection microscope can be used as a transmission microscope, but a full 4Pi configuration remains to be built. Zhou et al. proposed a similar concept in Ptychographic Diffraction Tomography, called opposite illumination [306], in which a transmission OTF (Figure 2c) is combined with a reflection OTF with wavelength variation ([63], Figure 10c), achieving quasi-isotropic resolution.

Another domain of possible progress remains the limited resolution, while fluorescence nanoscopy is now a reality, even commercially available, with routine resolution in the 50–100 nm range for biological applications, up to a few nanometers in some cases; if one does not want or cannot use fluorescence labeling, performances and versatility are way lower. The resolution of the far-field microscopy detection of evanescent waves through the scanning near-field optical microscopy (SNOM), being limited by the wave propagation, has been proposed [307], as well as the so-called superlenses [308] or simply microspheres [309]. However, all these approaches are based on near-field conversion/manipulation. Consequently, they are, by definition, limited to surface imaging and cannot deliver 3-D images of the interior of transparent samples. The development of far-field optical microscopy with superresolved images is now a very active field (see, for example, [310,311,312] and the references therein). In tomographic microscopy, most promising results have been obtained, up to date, in reflection configuration. Combining angular scanning and polarization [313] with advanced reconstruction algorithms taking into account multiscattering has enabled us to achieve a resolution of λ/10 in the far field [314]. Angular scanning and multispectral illumination also allow for more precise tomographic reconstructions [315]. It has also been demonstrated that confocal reconstructions are possible from tomographic acquisitions [127], delivering superior resolution images, albeit with intensity-only images, and not the index of refraction images. Note that the precise measurement of a resolution in coherent imaging requires some precautions [273,316]. The imaging of a sample’s fluctuations [264], apart from allowing us to study intracellular dynamics, also permits us to greatly improve resolution in tomographic imaging. Pump-probe approaches are also very promising [317], as temperature variations induce optical index variations, which are easily detected via the phase shift that the illumination beam then experiences [318].

On the opposite side of the scale, the imaging of large samples has also necessitated numerous works in optical tomographic microscopy. Cell cultures, spheroids, and animal models such as *C. elegans* worm or Zebrafish present specific difficulties because of their size and internal structures, inducing large phase retardation as well as multiple scattering properties. Single-pixel detection has also been adapted to imaging millimeter-scale samples [102,172,280,304]. When the sample is too thick and/or contains too much scattering, or exhibits a very high absorbance, coherence is lost, and one enters into the diffuse imaging domain, or, simply, no light emerges from the specimen. For such samples, tomography in reflection (epi-tomography) can, however, still deliver useful information. Promising results have already been obtained (see [319,320] and the references therein) and should encourage further developments, for example, speckle diffraction tomography [321].

Table 1 briefly recalls the main characteristics of this imaging technique, as well as some challenges it presently faces in order to improve performances, as well as to experience a wider adoption by the end users.

## 7. Conclusions

TDM is known to deliver high-quality 3D refractive index images of transparent or weakly absorbing samples, mainly encountered in biological imaging, in which many applications have already been developed. However, the technique is not limited to this field, and finds applications in material sciences, surface characterization, etc. However, one of the main limitations of TDM, compared, for example, to fluorescence microscopy or Coherent Anti-Stocks Raman Scattering (CARS) microscopy, is a lack of selectivity in the reconstructed images. As a matter of fact, several types of sample structures can exhibit the exact same refractive index. Therefore, bringing back selectivity to TDM images is of tremendous importance. For this purpose, several research groups are exploiting the vectorial nature of light for both acquisitions and reconstruction algorithms in order to exploit sensitivity to sample polarizability. This adds the possibility for TDM to characterize sample birefringence and to extract polarization information such as local retardance and polarization orientation in 3D. If chemical selectivity is sought, exploiting absorption information brought by TDM is of great help. Despite being often neglected, the wavelength dependence of the sample’s absorption can be beneficially used to help distinguishing several chemical species. Therefore, coupling between TDM and hyperspectral imaging has also been considered. Properly coupled with polarization, it could probably exhibit an even greater selectivity.

One can also focus on data reconstruction. Knowledge of the detailed image formation model of conventional microscopes coupled with tomographic reconstruction makes it possible to perform multimodal reconstructions of investigated samples. However, computational work is, up till now, still considered with either the first Born approximation or the Rytov approximation as an image formation model. This intrinsically limits the capabilities of TDM to tackle the imaging of multiple-scattering thick samples and strong refractive index variations. For this purpose, models relying on multi-layer approaches, coupled dipole resolution of Maxwell equations, or the Born series approximation of Maxwell equations have been considered. Moreover, coupled with an adequate reconstruction formalism or inverse approaches, these reconstruction methods are able to bring a solution to the “missing frequency” areas that are commonly occurring in 3D aperture synthesis.

The great development this field has experienced in the last 20 years [8,63,64,65,66,67,68,283,286,287] was enabled by the spectacular progress made in terms of speed, sensitivity, and decreasing costs of lasers, cameras, and computers. This review also identifies future tracks to be explored and hot topics to be addressed. These certainly leave TDM development open to a bright future, including the new capabilities enabled by deep-learning approaches, whether for hologram denoising, phase map computations, sample reconstructions, or specimen analysis [188,222,223,224,225,325,326,327,328,329,330,331,332,333,334,335].

## Figures and Tables

**Figure 1 sensors-24-01594-f001:**
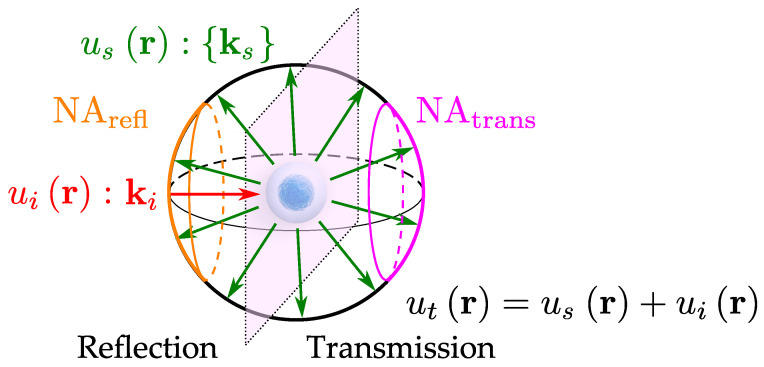
Geometrical interpretation of light diffraction within the first Born approximation.

**Figure 2 sensors-24-01594-f002:**
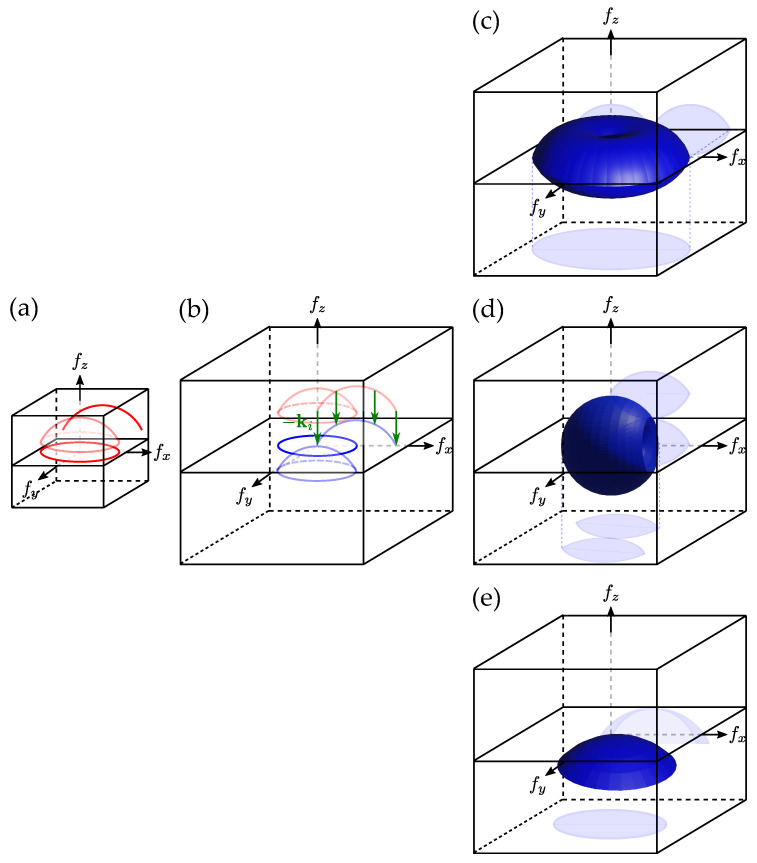
Illustration of the 3D synthetic aperture process. (**a**) Cap of the Ewald sphere collected. (**b**) Reallocation in a double-sized Fourier space. Aperture synthesis with (**c**) illumination angle variation, (**d**) rotation of the object, and (**e**) illumination wavelength variation.

**Figure 3 sensors-24-01594-f003:**
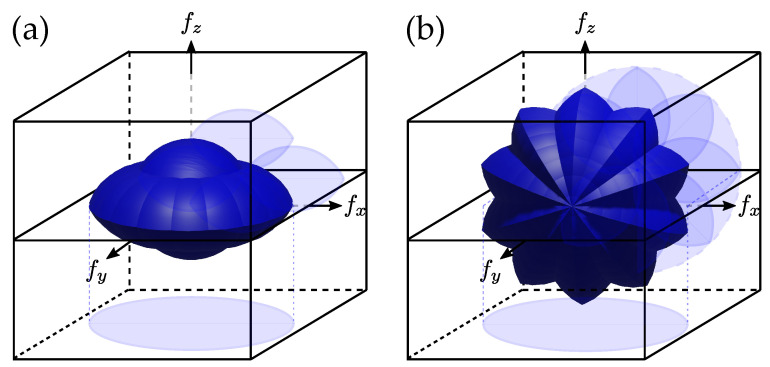
A 3D synthetic aperture process with combined approaches. OTF obtained with (**a**) a full rotation of the sample combined with one illumination sweep, (**b**) ann a full illumination scan for a few object rotation angles.

**Figure 4 sensors-24-01594-f004:**
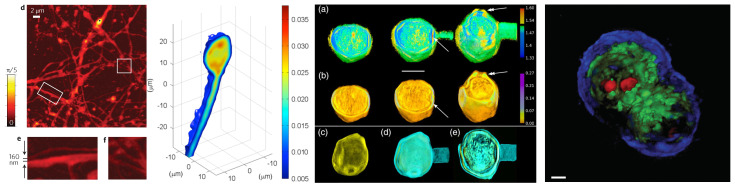
Examples of tomographic acquisitions with various systems. Far left: 3D aperture synthesis with illumination sweep, adapted from Figure 2 of Ref. [32]. Super-resolved phase obtained by so-called 2π-DHM reveals the spatial order of a self-assembled neural network: full-field (**d**) and magnifications of regions outlined by white squares (**e**,**f**). Left: 3D aperture synthesis with sample rotation. Figure 6 from Ref. [70] depicts the relative refractive index distribution in an HT-1080 cell with an extension and vertical cross sections through the cell. Refractive index peak-to-valley value Δn = 0.032 ± 0.004. Right: 3D aperture synthesis with combined sample rotation and illumination sweep. Figure 4 is from [37]. Betula pollen grain observed with TDM. Panels (**a**) and (**b**) show volumetric cuts (x–y views) through the 3D index of refraction image and absorption image, respectively. Note the higher index of refraction of the pollen walls, especially near the pores (double-headed arrow), and the double-layer outer wall (arrow). (**c**) Outer view of the pollen: image of the absorption component, displayed in yellow. (**d**) Outer view of the pollen: image of the complex index of refraction, with the refractive component displayed in cyan. (**e**) (x–y) Cut through the pollen. Scale bar: 10 μm. Far right: 3D aperture synthesis in white-light tomography. Adapted from Figure 4 of Ref. [59]. False-colour three-dimensional rendering of HT29 cells acquired in white-light tomography; z-stacks of 140 images. Scale bar 5 μm.

**Figure 5 sensors-24-01594-f005:**
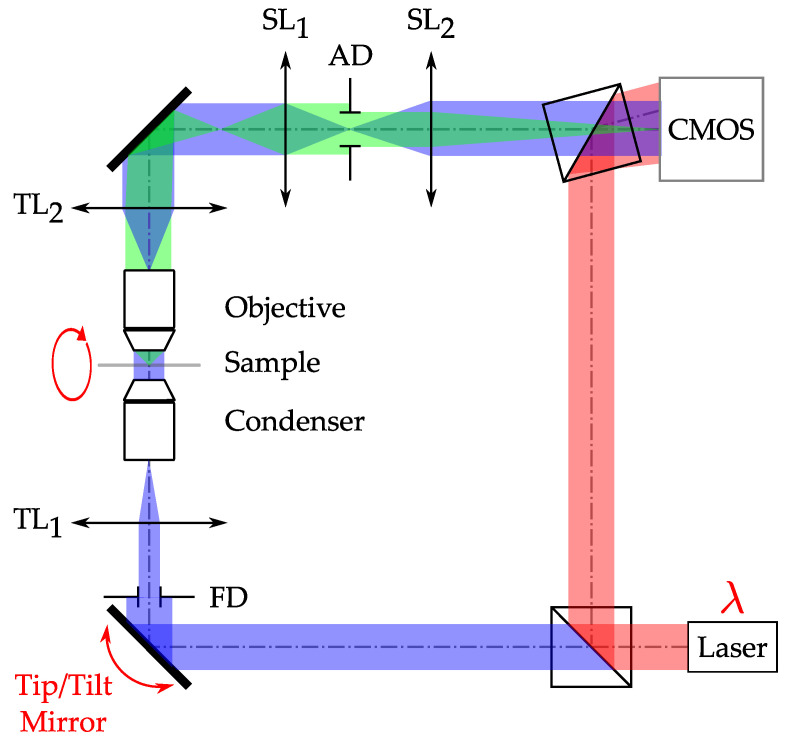
Sketch of a DHM configuration based on a Mach–Zehnder interferometer. FD: field diaphragm, AD: aperture diaphragm, TL: tube lens, SL: sampling lens. The red part is the part of the set-up that can be used to perform tomographic acquisitions.

**Figure 7 sensors-24-01594-f007:**
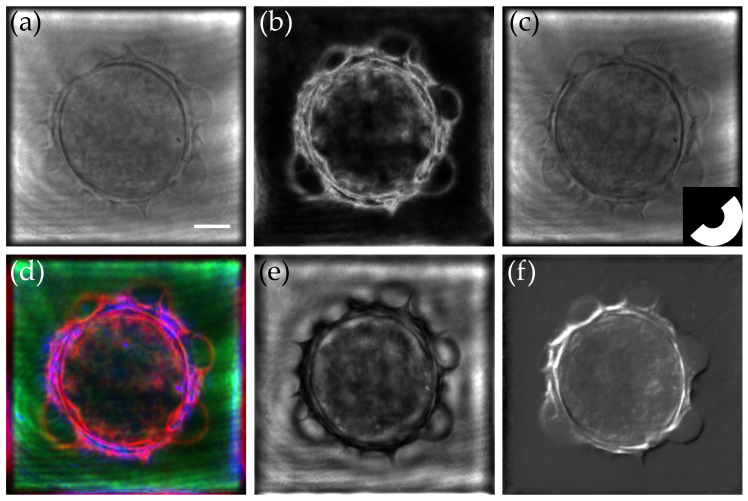
Simulated microscopy modalities. (**a**) Brightfield, (**b**) darkfield, (**c**) oblique illumination, (**d**) Rheinberg illumination, (**e**) negative phase contrast, and (**f**) DIC microscope. Scale bar is 10 μm. Adapted from Ref. [121].

**Figure 8 sensors-24-01594-f008:**
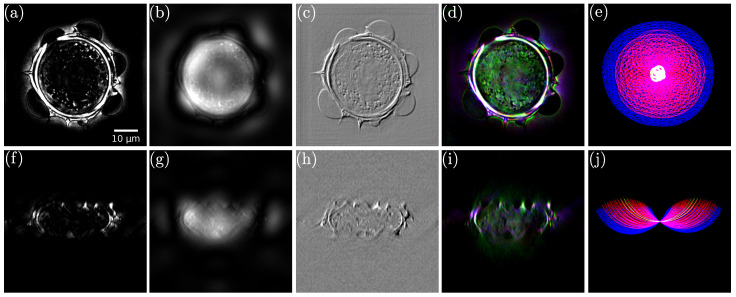
Simulated 3D microscopy modalities. (**a**–**e**) Lateral cuts, (**f**–**j**) axial cuts. (**a**,**f**) darkfield, (**b**,**g**) positive phase contrast, (**c**,**h**) DIC microscope, (**d**,**i**) Rheinberg Illumination, and (**e**,**j**) Rheinberg spectrum. Scale bar is 10 μm. Adapted from Ref. [121].

**Figure 9 sensors-24-01594-f009:**
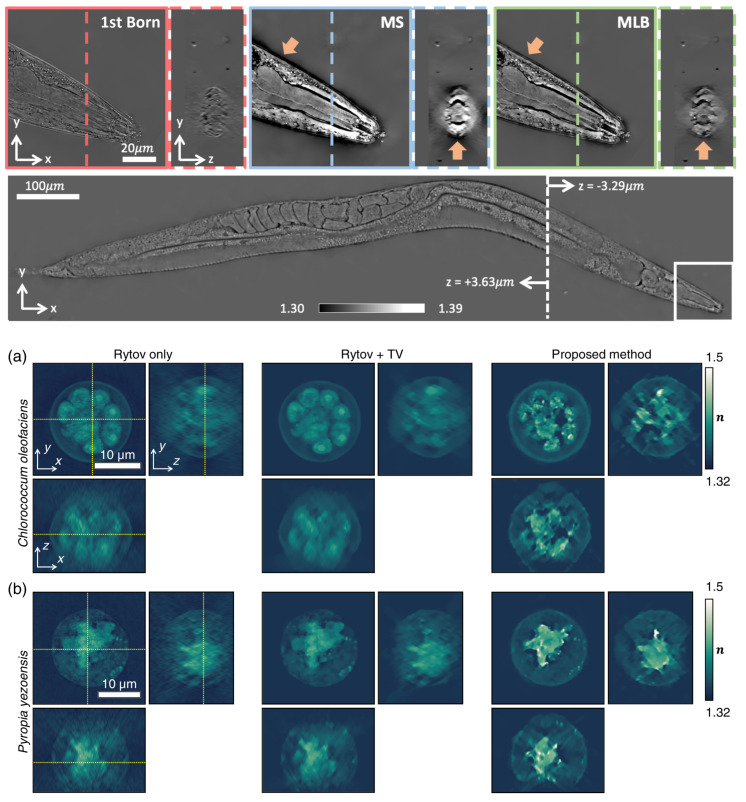
Top: Figure 6 from Ref. [145]: 3D refractive index reconstructions of an entire adult hermaphrodite *C. elegans* worm from the dataset of Ref. [148], using MLB model. The insets show zoomed-in comparison between orthonormal cross-sections using the first Born, multi-slice (MS), and MLB methods, for the white box region that includes the mouth and pharynx of the *C. elegans*. Bottom: Adapted from Figure 4 from Ref. [149]: reconstruction results of optically thick specimens: (**a**) *Chlorococcum oleofaciens* and (**b**) *Pyropia yezoensis*. Each column represents cross-sectional images reconstructed with the Rytov approximation (first column), Rytov approximation with total variation regularization (second column), and the new method proposed in [149] (third column).

**Figure 10 sensors-24-01594-f010:**
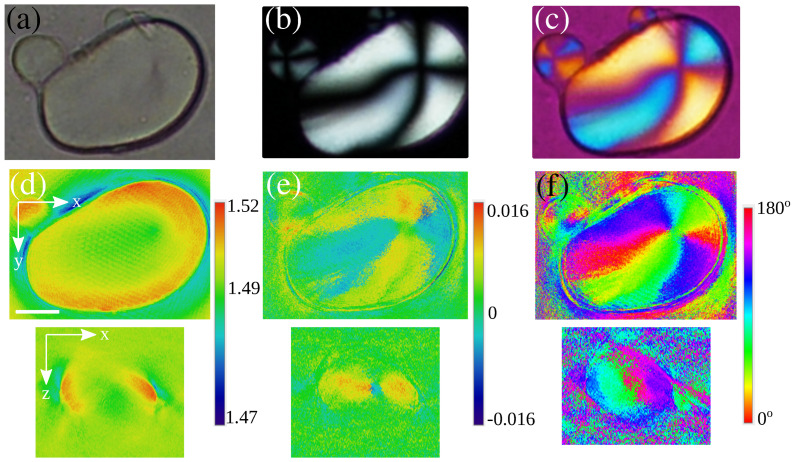
Polarization-resolved acquisition of a potato starch. (**a**) Brightfield image. (**b**) Polarized light microscope (PLM) image. (**c**) PLM & phase plate image. (**d**) Refractive index map. (**e**) Birefringence map. (**f**) Polarization orientation map. Adapted from Ref. [73].

**Figure 11 sensors-24-01594-f011:**
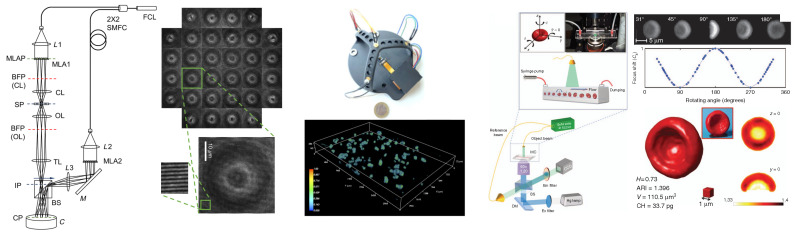
Examples of simplified tomographic setups. **Left**: snapshot tomography, adapted from Figures 2 and 3 of Ref. [193]. Schematic diagram of the experimental setup: FCL, fiber-coupled laser; 2 × 2 SMFC, 2 × 2 single-mode fiber coupler; L1, L2, and L3, lenses; MLA1 and MLA2, microlens arrays; CL, condenser lens; OL, objective lens; TL, tube lens; BS, beam splitter; M, mirror; C, camera; MLAP, MLA plane; BFP, back focal plane; SP, sample plane; IP, image plane; CP, camera plane. The raw interferogram image of Henrietta Lacks (HeLa) human-cervical-cancer cells consists of multiple projection images, one of which is enlarged on the right. **Middle**: lens-free tomographic microscope, adapted from Figures 1 and 7 of Ref. [240]. Setup using only 4 illuminations and a CMOS image sensor and 3D reconstruction of intestinal organoids embedded in Matrigel over a volume of more than 3.4 mm × 2.3 mm × 0.3 mm. Color bar: normalized scattering potential. **Right**: tomographic flow microscopy, adapted from Figures 1 and 2 of Ref. [232]. Sketch of the experimental R-TPM set-up. Cells are injected into a microfluidic channel and tumble while flowing along the y-axis. At the same time, a holographic image sequence is acquired. In the top-left corner of the inset, the reference system for cell tumbling is reported; in the top-right corner, a photo of the real set-up is shown. Rotation occurs around the x and z axes. BS, beam splitter; DM, dichroic mirror; MC, microchannel. Images of R-TPM, applied on RBCs presenting one-side-concavity morphological anomalies (Ho0.9) with respect to the ideal healthy one. QPIs and mathematical dependence of the defocus coefficient from the rotation angle and the tomogram retrieved by the QPIs and the RI distributions at the z = 0 and y = 0 planes. ARI, V, and CH are also reported together with the plastic 3D representations realized with a 3D printer.

**Figure 12 sensors-24-01594-f012:**
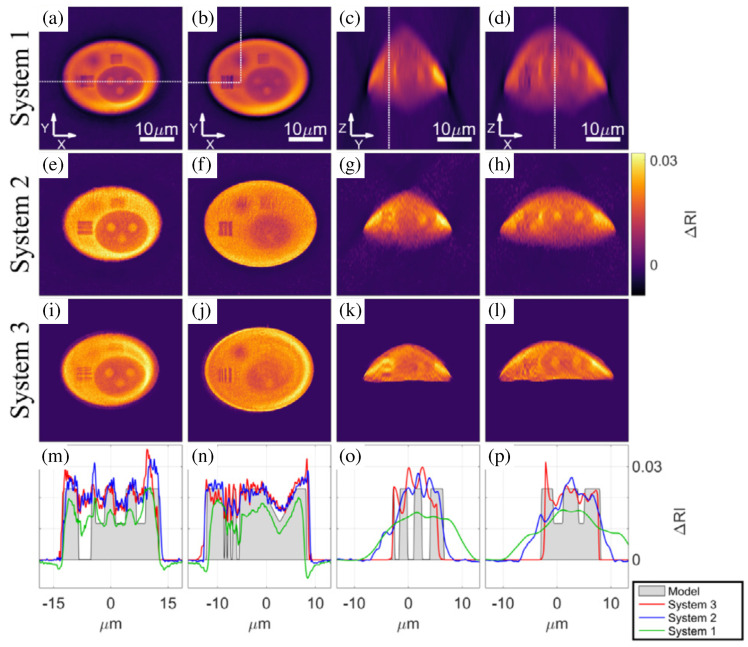
Use of an artificial cell for reconstruction accuracy assessment. We are referencing Figure 8 from Ref. [284]: Cross-sections of the 3D RI distribution of the cell phantom measured experimentally in 3 LA ODT systems: (**a**–**d**) System 1, (**e**–**h**) System 2, (**i**–**l**) System 3. (**m**–**p**) RI profiles along the white dotted lines indicated on the corresponding panels (**a**–**d**).

**Table 1 sensors-24-01594-t001:** Main characteristics of tomographic microscopy and present challenges.

	Field of View	Lateral Resolution	Longitudinal Resolution	Acquisition Speed	Refractive Index Sensitivity	Cost
Present situation	From about 100 × 100 μm at high-resolution to millimeter-size samples [240,280,322]	Sub 100 nm lateral resolution demonstrated [32,37]	Usually 2–3 times lower than lateral resolution. Isotropic resolution of about 180 nm demonstrated [37]	From a few seconds (e.g., rotating arm scanning [38,86]) to camera-speed limit only [193]	Δn = $10^{-2}$ commonly obtained, even on biological samples, up to Δn = $4.21\times 10^{-5}$ in large plastic samples [240]	From a few 100€ (lensless tomography) to about 50–70k€ (estimated) for a high-end 4Pi system (salaries and computers not included)
Challenges	Keeping high resolution in large volumes as in light-sheet microscopy	Development of nanometric super-resolution for unlabeled samples [311,312]	Isotropic resolution using standard configurations	High resolution with rapid acquisitions compatibility. Combination with other microscopic techniques [323]	Development and adoption of metrological approaches [273,282,283,284]	Development of open-source alternatives, such as the OpenSPIM initiative [324]

## Data Availability

Data underlying the results presented in this paper are not publicly available at this time but may be obtained from the authors upon reasonable request.

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
