# Peer review of "Recent Advances and Current Trends in Transmission Tomographic Diffraction Microscopy"

_sensors, 2024, doi:10.3390/s24051594_

Round 1

Reviewer 1 Report

Comments and Suggestions for Authors

The manuscript “Recent advances, and current trends in transmission tomographic

diffraction microscopy” by Nicolas Verrier et al. has reviewed the history, development and future of tomographic diffraction microscopy (TDM). The basic concepts and principles of TDM are elucidated in the manuscript. Later on, the authors have presented the system setup and reconstruction algorithms to achieve the biomedical images.

Overall, I would like to suggest this manuscript to be published after minor corrections. My suggestions are as follows:

1) It is suggested to put more keywords in this manuscript to guide readers;

2) In the configuration of DHM system setup in Fig.4, it is sort of confusing that if the condenser and objective are the same. Also, please double-check the light ray for the blue and green color, I am not quite sure why we should put the TL1 in the system?

3) In Fig.6 and Fig.7, the length of the scale bar should be presented in the caption;

4) When the authors mention that the images are simulated. Could you please give the process on how to simulate all these images for darkfield,PC and so on.

5) The caption of 6.2, word of “Fonctionalization” is a typo.

Author Response

Comment to Reviewer 1
The manuscript “Recent advances, and current trends in transmission tomographic diffractionmicroscopy” by Nicolas Verrier et al. has reviewed the history, development and future of tomographic diffraction microscopy (TDM). The basic concepts and principles of TDM are elucidated in the manuscript. Later on, the authors have presented the system setup and reconstruction algorithms to achieve the biomedical images.
First of all, we would like to thank the Reviewer for the positive remarks. We hope we will be able to give satisfactory answers to the Reviewers’ concerns. Suggestions have been accounted for, and emphasized by red font in the resubmitted manuscript.

Overall, I would like to suggest this manuscript to be published after minor corrections. My suggestions are as follows:
1) It is suggested to put more keywords in this manuscript to guide readers;
We added the following keywords: polarimetric/vectorial imaging; multiple scattering; Holography; Diffraction; Fourier Optics

2) In the configuration of DHM system setup in Fig.4, it is sort of confusing that if the condenser and objective are the same. Also, please double-check the light ray for the blue and green color, I am not quite sure why we should put the TL1 in the system?
We double checked the light path, and it is right. It should be kept in mind that both objective and condenser are infinite corrected, thus a tube lens should be added for image formation. TL1 is here to ensure that the sample is illuminated under collimated light. To clarify this point, the following paragraph has been added line 185:
“Please note that in Fig. 4, the condenser is a microscope objective. Here, TL1 grants light collimation on the investigated sample. The same can equally be done with a more conventional condenser, as long as the object remains illuminated by a plane wave.”

3) In Fig.6 and Fig.7, the length of the scale bar should be presented in the caption;
“Scale bar is 10 μm” has been added in both captions

4) When the authors mention that the images are simulated. Could you please give the process on how to simulate all these images for darkfield, PC and so on.
Details about the simulation has already been published in another journal. A general scheme is proposed Fig. 5. The simulation codes are available in our Github, whose link has been added in the manuscript. We also added the following sentence:
“More details about the simulation process can be found in Ref. [ 113 ]. Python version of our simulation and tomographic reconstruction codes can be found in Github [120].”

5) The caption of 6.2, word of “Fonctionalization” is a typo.
Typo has been corrected.

Full response Letter is enclosed for complete view of our answers to both reviewers.

Reviewer 2 Report

Comments and Suggestions for Authors

This review article deals with the topic of transmission tomographic diffraction microscopy. The topic is well introduced even for reader less versed in the topic, which is followed by a thorough covering of the theoretical background. On this basis, the current state-of-the art, hot topics and future perspectives are reviewed, with plentiful citations to works of interest. Reconstruction of microscopic images is also demonstrated on the basis of previous work of the authors.

The review is very well written, it introduces and covers the topic in a way review article should. I can recommend the article to be published, and offer only a few minor and optional suggestions:

* Applications: although many applications of the method are mentioned during the text, a dedicated chapter reviewing the most important and potential applications would be beneficial.

* Clarity of text:

- line 62: Combining both tomographic approaches: it is not clear to me from preceding text which two tomographic approaches?

- l.113: lambda_0: I might have missed it, but I cannot find where it is defined

* Language and editing:

- line 20: down to a few nanometer -> down to a few nanometers

- l.82: “: The total field “ extra :

-l.86, elsewhere in the text: “Resolution” of Equation. “Solution” of equation sounds more correct to me.

- l.284, 393: “are proposed Fig.” -> “are proposed in Fig.“, in general, “shown” in figure sounds more correct to me.

- l.572: “, so the phase change the illumination beam experiences” not understandable, maybe “, so the phase changes the illumination that the beam experiences”?

- l.583: “does not limit to this field, “ -> “is not limited to this field, “ ?

- l.583: “and find applications in “ -> “and finds applications in “ ?

- l.590: “such as local retardance, polarization orientation in 3D” -> “such as local retardance, polarization, and orientation in 3D” ?

Comments on the Quality of English Language

With exception of the few typos mentioned above, the English is of high quality and very understandable.

Author Response

Comment to Reviewer 2
This review article deals with the topic of transmission tomographic diffraction microscopy. The topic is well introduced even for reader less versed in the topic, which is followed by a thorough covering of the theoretical background. On this basis, the current state-of-the art, hot topics and future perspectives are reviewed, with plentiful citations to works of interest. Reconstruction of microscopic images is also demonstrated on the basis of previous work of the authors.
The review is very well written, it introduces and covers the topic in a way review article should. I can recommend the article to be published, and offer only a few minor and optional suggestions:
We do thank the review for the very encouraging remark. Suggestions have been accounted for, and emphasized by blue font in the resubmitted manuscript.

* Applications: although many applications of the method are mentioned during the text, a dedicated chapter reviewing the most important and potential applications would be beneficial.
A new section “6.4. Promising applications” has been added to the originally submitted Manuscript.

* Clarity of text:
- line 62: Combining both tomographic approaches: it is not clear to me from preceding text which two tomographic approaches?
We modified this sentence to : “Combining both illumination scanning and sample rotation [...]”
- l.113: lambda_0: I might have missed it, but I cannot find where it is defined
The variable should have been read $\lambda_v$. We do apologize for the typo.

* Language and editing:
- line 20: down to a few nanometer -> down to a few nanometers
- l.82: “: The total field “ extra :
-l.86, elsewhere in the text: “Resolution” of Equation. “Solution” of equation sounds more correct to me.
- l.284, 393: “are proposed Fig.” -> “are proposed in Fig.“, in general, “shown” in figure sounds more
correct to me.
- l.572: “, so the phase change the illumination beam experiences” not understandable, maybe “, so the
phase changes the illumination that the beam experiences”?
- l.583: “does not limit to this field, “ -> “is not limited to this field, “ ?
- l.583: “and find applications in “ -> “and finds applications in “ ?
- l.590: “such as local retardance, polarization orientation in 3D” -> “such as local retardance,
polarization, and orientation in 3D” ?
All the suggestion have been accounted for.

Please find enclosed the full answer letter for information.
